# Simulated operant reflex conditioning environment reveals effects of feedback parameters

Kyoungsoon Kim[1], Ethan Oblak[2], Kathleen Manella[3], James Sulzer[4]*

1 University of Texas at Austin, Austin, Texas, United States of America, 2 RIKEN Center for Brain Science, Saitama, Japan, 3 Nova Southeastern University, Clearwater, Florida, United States of America, 4 MetroHealth Hospital and Case Western Reserve University, Cleveland, Ohio, United States of America

* jss280@case.edu

**Data Availability Statement:** Data is available at: Sulzer, James, and Kyoungsoon Kim. 2023. "Operant Conditioning Simulation Paradigm." OSF. May 26. DOI 10.17605/OSF.IO/Y8P47 https://osf.io/y8p47/.

## Abstract

Operant conditioning of neural activation has been researched for decades in humans and animals. Many theories suggest two parallel learning processes, implicit and explicit. The degree to which feedback affects these processes individually remains to be fully understood and may contribute to a large percentage of non-learners. Our goal is to determine the explicit decision-making processes in response to feedback representing an operant conditioning environment. We developed a simulated operant conditioning environment based on a feedback model of spinal reflex excitability, one of the simplest forms of neural operant conditioning. We isolated the perception of the feedback signal from self-regulation of an explicit unskilled visuomotor task, enabling us to quantitatively examine feedback strategy. Our hypothesis was that feedback type, biological variability, and reward threshold affect operant conditioning performance and operant strategy. Healthy individuals (N = 41) were instructed to play a web application game using keyboard inputs to rotate a virtual knob representative of an operant strategy. The goal was to align the knob with a hidden target. Participants were asked to "down-condition" the amplitude of the virtual feedback signal, which was achieved by placing the knob as close as possible to the hidden target. We varied feedback type (knowledge of performance, knowledge of results), biological variability (low, high), and reward threshold (easy, moderate, difficult) in a factorial design. Parameters were extracted from real operant conditioning data. Our main outcomes were the feedback signal amplitude (performance) and the mean change in dial position (operant strategy). We observed that performance was modulated by variability, while operant strategy was modulated by feedback type. These results show complex relations between fundamental feedback parameters and provide the principles for optimizing neural operant conditioning for non-responders.

## Introduction

Operant conditioning is a commonly used procedure that provides a reinforcing stimulus as a consequence to a desired behavior [1]. Operant conditioning of neural activity has been

**Funding:** This work was financially supported in part by the NIH/NICHD (P2CHD086844, Kautz), and JS is the recipient. This work was also supported by the NICHD under the National Institutes of Health under the Award Number R01HD100416. The contents are solely the responsibility of the authors and do not necessarily represent the official views of the NIH or NICHD. The funders had no role in study design, data collection and analysis, decision to publish, or preparation of the manuscript.

**Competing interests:** The authors have declared that no competing interests exist.

investigated for decades [2, 3], primarily at the level of the brain [4]. In this manner, an individual learns to self-modulate a neural circuit, with the implication of neuroplastic changes [4–7]. LaCroix [8] hypothesized that in operant conditioning there are both implicit automatic learning processes occurring in parallel with explicit ones. However, these processes combined with physiology of a complex brain circuit is a poorly understood process [4]. As opposed to complex neural circuity, there exists a simpler model for neurofeedback that targets well-understood monosynaptic stretch reflexes [9–11]. In this model, an evoked response from electrical stimulation of the peripheral nerve is measured from the innervated muscle (i.e. H-reflex [12]), and its amplitude is directly provided as feedback to the user. This technique, known as operant H-reflex conditioning, was developed by Wolpaw and colleagues both in human and animal models [13, 14]. These researchers have shown the importance of corticospinal tracts in enabling operant learning of the H-reflex amplitude [15], indicated the sites of neuroplastic changes on a synaptic level [16], and provided evidence for its translation to humans [14]. They suggest that operant H-reflex conditioning provides an excellent model for learning of a simple motor skill [13, 17].

Operant H-reflex conditioning typically requires about three months of training. However, there are a substantial portion of individuals who do not learn to self-modulate the spinal circuit, known as non-responders [4, 6]. Such an investment in training for an unknown outcome becomes prohibitive for both scientific research and clinical translation, thus it is critical to understand why certain individuals have difficulty in learning. Operant H-reflex conditioning has traditionally been assumed to rely on implicit mechanisms because of its initiation in rodent models [9, 18, 19]. Further, explicit self-modulation strategies reported in earlier operant H-reflex conditioning work were unrelated to CNS processes responsible for the task-dependent changes in H-reflex size [14]. In contrast, functional magnetic resonance imaging (fMRI) neurofeedback was initiated in humans and was first believed to be entirely governed by explicit processes [7]. However, a contemporary work showed implicit learning with no instructed explicit strategy was successful in modulating early visual cortical activity [20], and further evidence solidified that implicit learning mechanisms were indeed feasible [21]. The relative roles of implicit and explicit processes in operant conditioning of neural signals is still debated [4, 20, 22], and may be influenced by the trained neural substrate. Our anecdotal evidence suggests that even though prior to training we have no conscious ability to regulate monosynaptic reflex activity, explicit mechanisms may play a role in operant H-reflex conditioning [23]. Specifically, we observed the performance of a post-stroke individual trained to operantly condition the H-reflex of the rectus femoris (RF) remained static until the participant was consistently reminded of the instructions.

We have developed a paradigm that helps isolate explicit and implicit learning mechanisms in neural operant conditioning. We previously developed a simulated fMRI neurofeedback environment that separated the ability to self-regulate the neurofeedback signal from its perception by using an explicit, unskilled visuomotor task [24]. We then validated this model empirically in a fMRI neurofeedback experiment [25]. By creating artificial brain activity based on real data and testing outside the scanner, we could vary certain parameters such as hemodynamic delay and feedback delay to efficiently examine the effects of feedback on learning. We found that the feedback timing and hemodynamics affected strategy and performance. These results critically emphasize the importance of understanding the characteristics of the feedback signal that modulate explicit strategy and ultimately operant conditioning performance.

The goal of this study was to further understand the explicit aspect of learning during the operant H-reflex conditioning process. As in our previous simulated neurofeedback paradigm [24], operant conditioning can be broken down into perception, decision making,

conditioning ability, and biological variability (noise) components. Based on our data of operant H-reflex conditioning [23], we can simulate the level to which individuals can condition the H-reflex, as well as the range of variability of the H-reflex. Assuming unimpeded perception, we can isolate the effects of simulated feedback on explicit decision-making processes likely occurring during actual neural operant conditioning. Our hypothesis was that several factors likely affected explicit learning: 1) *Kr* feedback will improve performance compared to *Kp* feedback [26], 2) larger biological variability will result in worse performance, and 3) the reward threshold modulates performance and strategy.

During this cognitive experiment, neurologically intact participants were asked to play a simple computer game, wherein the task was to rotate a virtual rotary knob representative of an operant strategy. The simulated H-reflex was composed of a knob-controllable canonical form of the H-reflex with biological variability and measurement signal noise extracted from the actual H-reflex data collected previously [23]. As the inherently variable nature of H-reflex amplitude has been shown to be correlated with the fluctuations in motoneuronal membrane potential [27], application of data-driven biological variability and signal noise enabled our simulation environment to be as similar to the neurophysiological environment of operant H-reflex conditioning as possible. The goal was to align the knob with a hidden target, of which the proximity of the knob to the target was proportional to the feedback signal. In other words, participants were asked to "down-condition" the virtual feedback amplitude, which was achieved by placing the knob as close as possible to the hidden target. The task is analogous to a tuning dial on a radio, where the participant aims to find the best signal. Thus, we are replacing a skilled operant conditioning task with a simplified, unskilled tuning task that enables us to quantify operant strategies at an unprecedented level. We then compared this model to real operant H-reflex conditioning performance to demonstrate the similarity in learning processes between the simulation environment and that of operant H-reflex conditioning. This experiment represents a novel approach to understanding the role of feedback parameters in explicit learning that is likely taking place during neural operant conditioning. The simulated operant H-reflex conditioning environment provides an efficient method for analyzing and possibly improving learning. This approach may also assist in identifying non-responders, ultimately enhancing the robustness of operant H-reflex conditioning.

## Methods

### Subjects

A total of 41 healthy participants were recruited with no history of vision impairment, cognitive impairment, and neurological disease or injury. The participants were 24 men and 17 women aged 20–41 between April and July 2021. The study was considered exempt by the University of Texas at Austin Institutional Review Board. While consent was not required, it was acquired via email or verbally from all participants prior to participation. There was no documentation collected linking participant identity to their data. To prevent variation in one's concentration level, sessions were performed at the same time of the day for everyone. The experiment consisted of three sessions, Sessions 1 and 2 took approximately 1.5 hours to complete and Session 3 took approximately 30 minutes. All sessions were performed at least a day apart.

### Experimental protocol and data collection

This experiment was conducted virtually, where the participant was asked to use their personal computer and play a simple computer game via a web application (web app). The web app was designed using the Matlab App Designer (MathWorks, Natick, MA), and was established at

the University of Texas at Austin's Matlab Web App server. Participants were emailed a link prior to the experiment. The participant and investigator met virtually on the scheduled date, where a webcam was used to monitor the session to ensure attention and efficient trouble-shooting in case of any problems. The experimental protocol consisted of 4 stages: instruction, consent, demonstration (practice run), and the experiment. The experiment was carried out over three separate sessions at least a day apart.

During the Instruction stage, participants were briefly informed about the purpose of the study, approximate duration, future experimental schedule, and compensation for their participation. The virtual consent was held as a reconfirmation of the participants' voluntary commitment toward the task and individuals were free to withdraw from the experiment whenever desired. If the participant chose to participate in the study, the participant provided their full name and e-mail address as an electrical signature. Personal information acquired at this stage was encrypted and data was anonymized based on this encryption.

The Demonstration stage was considered a practice run and participants were informed of the visual layout, the keyboard control, task goal, and the experimental structure of the experiment in a detailed manner. The visual layout consisted of rotary dial knob, feedback bar, attempt number, and cumulative success rate (**Fig 1**). The layout was modeled after the traditional protocol in earlier work [14]. The rotation of the rotary knob was controlled by keyboard inputs, arranged based on desired hand: fast counterclockwise (A if left side desired or L if right side desired), slow counterclockwise (S, K), fast clockwise (F, H), slow clockwise (D, J), and select (spacebar). The task goal was to align the rotary knob with a hidden target (invisible during experiment but shown in orange in **Fig 1**). The displacement of the rotary knob is a one-dimensional reduction serving as a proxy for the decision making occurring during operant H-reflex conditioning. That is, the amount of angular displacement of the rotary knob is reflective of the mental strategy selection or effort during real operant H-reflex conditioning. The dial was programed to move after each keystroke and did not move continuously when the key was pressed and held. There was no time limit for each trial, and participants were

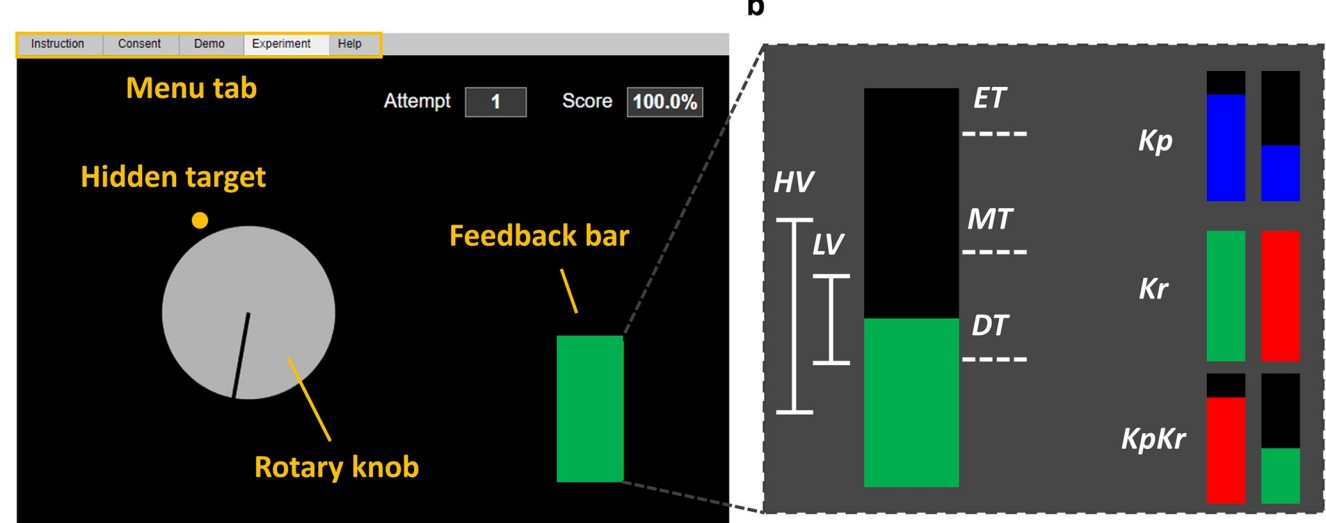

**Fig 1. Visual layout of the web application and feedback parameter. (a)** A feedback screen for the participant, in which they were asked to rotate the virtual rotary knob to find the hidden target. The feedback bar indicated the amount of error (Knowledge of Performance, *Kp* feedback) or success/failure information (Knowledge of Results, *Kr* feedback) in finding the target via changing the bar height (*Kp*) or color (*Kr*). Participants received additional feedback of a running score of their performance and trial number. **(b)** Visualization of feedback parameters: feedback type (performance, *Kp*, knowledge *Kr*, and both *KpKr*), biological variability (low, *LV* and high, *HV*), and reward threshold (easy, *ET*, moderate, *MT*, and difficult, *DT*).

allowed to move the dial as much as necessary, before confirming their decision by pressing the spacebar. Once the position was selected the web app generated a simulated H-reflex.

## Simulated H-reflex

We modeled the H-reflex (**Fig 2**) by decomposing it into parts to investigate the role of these subcomponents on learning in a simulation. For each trial (k), when the participant explored the hidden target and confirmed decision by pressing spacebar, a simulated H-reflex ($h_{sim}(k)$) was generated and presented as feedback bar graph (**Fig 3**).

The simulated H-reflex was composed of knob-controllable canonical form of the H-reflex, decision gain, biological variability and signal noise driven from the actual H-reflex data. The canonical form was a single period sine function. The amplitude of the sine function $a_c(k)$ was determined by the difference (error) between the dial position ($\theta(k)$) and the hidden target ($\theta_{tar}$), and multiplied by a predetermined gain, $g$:

$$a_c(k) = g(\theta(k) - \theta_{tar}) \tag{1}$$

The peak-to-peak value of the canonical form was a linear function set at 1 when the error was 180˚ and 0.5 when the error was 0˚. Thus, for down-conditioning, the best possible performance of 0.5 was extracted empirically from earlier work on operant down-conditioning of the RF H-reflex [23]. In other words, having a peak-to-peak value closer to 0.5 implied the

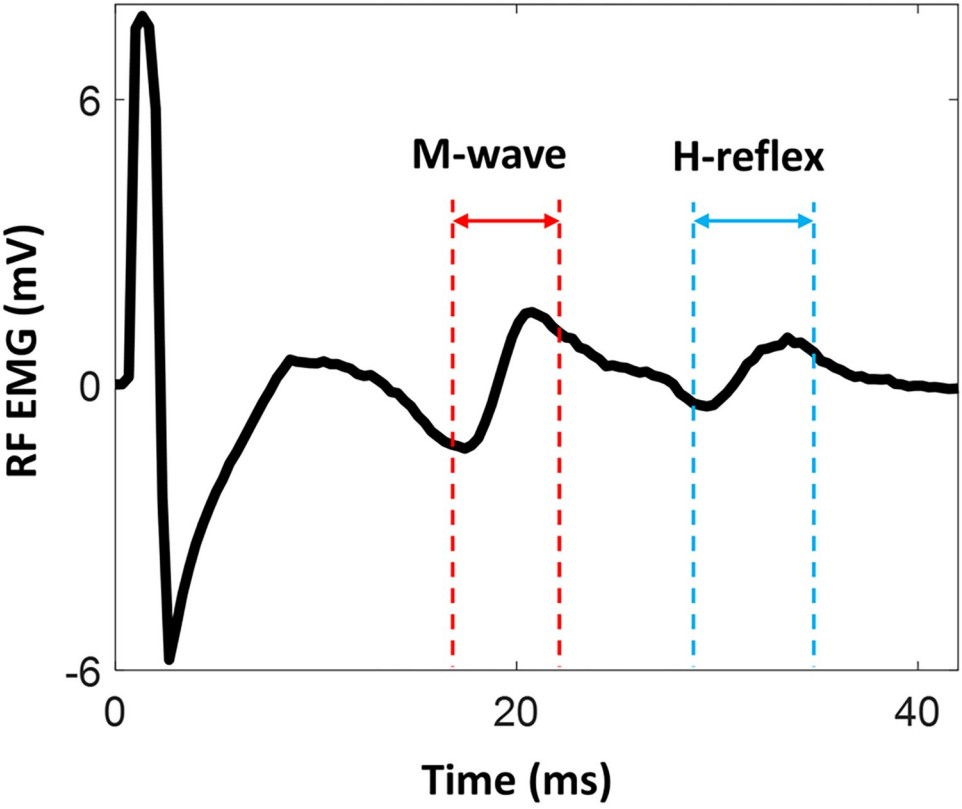

**Fig 2. Representation of rectus femoris (RF) H-reflex.** Electromyography (EMG) activity during femoral nerve stimulation is depicted. After the onset of electrical stimulation on the femoral nerve, a motor response (M-wave) and monosynaptic spinal reflex (H-reflex) is elicited. The H-reflex profile was decomposed to investigate the role of these subcomponents on learning in a simulation.

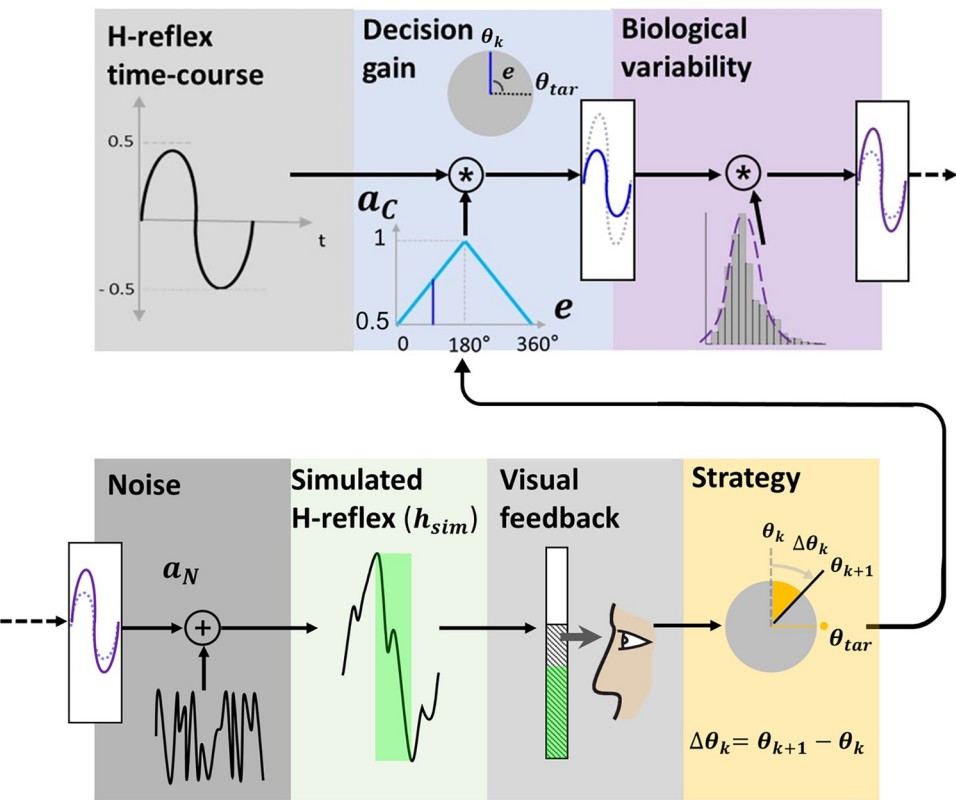

**Fig 3. Simulated H-reflex.** Simulated H-reflex ($h_{sim}$) is generated by applying decision gain ($a_c$), biological variability ($\beta$), and noise ($a_N$) on the H-reflex time-course and the peak-to-peak magnitude is provided as the visual feedback for the participant. Participant uses information to adjust one's strategy ($\Delta\theta$) to either minimize the feedback bar height or turn the bar green.

subject was successful in matching the hidden target, which can be translated as success in the context of down-conditioning, whereas value closer to 1 implied failure. We modeled the natural biological variability of the H-reflex ($\sigma^2$) with a normal distribution obtained from a set of participants' H-reflexes, also from previous work [23]. All analyses and distribution model fitting was performed using MATLAB software (MATLAB 2019a, Mathworks, Natick, MA). We used the minimum and maximum variability (**LV**, $\sigma^2$ = 0.25 and **HV**, $\sigma^2$ = 0.75, respectively) of the dataset. This variability was applied in the form of normal distribution ($\beta(k){\sim}N(\mu,\sigma^2)$) generated by Box-Muller transform [28]. Based on the same distribution, the dominant noise power ($a_N$) of H-reflex signals was analyzed using Fast Fourier Transform (FFT) and applied to the simulated H-reflex. To measure performance, we acquired the simulated H-reflex, $h_{sim}(k)$, using the following equation:

$$h_{sim}(k) = (\sin(t) * a_c(k) * \beta(k) + a_N)_{pk-pk} \tag{2}$$

For the Demonstration, the participant conducted 5 familiarization trials for two conditions, **Kp**, and **Kr**. Both terms are commonly used in psychology [26], where **Kp** provides information about how the task was achieved and **Kr** is knowledge about whether the goal of the task was achieved. In this sense, **Kp** focused on reducing the feedback bar height and **Kr** focused on maintaining the bar height below a threshold. During **Kp**, a blue bar was presented (**Fig 1B, top right**), in which the height was peak-to-peak magnitude of the simulated H-reflex. No bar color change was associated with **Kp** feedback. For **Kr**, the bar color changed based on

**a**

| Condition | Feedback | Biological variability | Threshold |
|:---:|:---:|:---:|:---:|
| C1 | *Kp* | *LV* | - |
| C2 | *Kp* | *HV* | - |
| C3 | *Kr* | *LV* | *MT* |
| C4 | *Kr* | *HV* | *MT* |
| C5 | *Kr* | *LV* | *DT* |
| C6 | *Kr* | *LV* | *ET* |
| C7 | *Kr* | *HV* | *DT* |
| C8 | *Kr* | *HV* | *ET* |
| C9 | *KpKr* | *LV* | *MT* |
| C10 | *KpKr* | *HV* | *MT* |

**b**

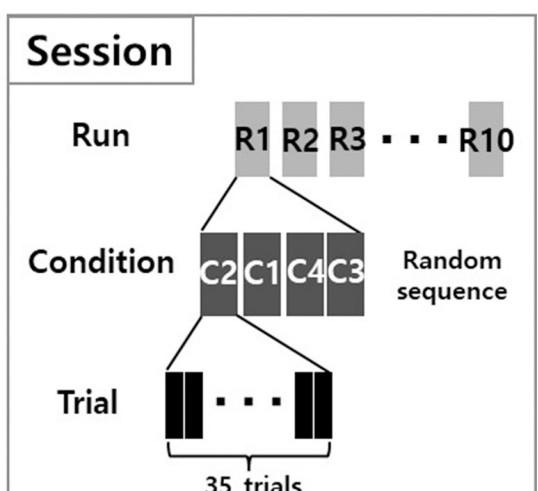

**Fig 4. Experimental condition and protocol. (a)** Total of ten conditions were tested, based on the different feedback types (performance, *Kp*, results, *Kr*, and both, *KpKr*), biological variability (low, *LV* and high, *HV*), and reward threshold (easy, *ET*, moderate, *MT*, and difficult, *DT*). **(b)** There were three sessions in this experiment. Each session was comprised of 10 runs. Within each run, there were 4 conditions for Session 1 and 2, and Session 3 had two conditions. For each condition, the participant was given 35 trials.

the result of the comparison between the simulated H-reflex magnitude and the pre-determined threshold levels (**Fig 1B, middle right**). The threshold levels were chosen as the 44th, 66th, and 77th percentile of the 1,350 evoked H-reflexes without feedback [23], designated as easy/moderate/difficult thresholds (***ET*** = 0.46, ***MT*** = 0.7 and ***DT*** = 0.96, respectively). If the magnitude of the simulated H-reflex was below the threshold, the bar turned green indicating success and the cumulative score increased. If $h_{sim}(k)$ was larger than the threshold, the bar turned red, indicating failure and the score decreased. The bar height and the threshold were kept hidden from the participants during the ***Kr*** condition and only the binary result with fixed bar height was provided as feedback. Not introduced in the Demonstration, but used later in the Experiment, is the combination of ***Kp*** and ***Kr*** feedback, which is the ***KpKr*** condition, but with a moving bar height as described above (**Fig 1B, bottom right**), reflecting current operant H-reflex conditioning practices [14].

Following the Demonstration, we introduced the Experiment phase. The experimental structure comprised of 10 different conditions including factors of three feedback types (***Kp/Kr/KpKr***), low/high biological variability (***LV/HV***), and three reward threshold levels (***ET/MT/DT***), the latter factor during ***Kr*** feedback only (**Fig 4A**).

We conducted 3 sessions on different days at least one day apart: Session 1 (C1-C4), Session 2 (C5-C8), and Session 3 (C9-C10) (**Fig 4B**). The Experiment stage consisted of 10 runs of each condition of 35 trials each (**Fig 4B**). A "trial" was defined as a single decision, where the individual explored the hidden target using keyboard inputs, confirmed one's decision by pressing spacebar, and feedback bar was updated accordingly. The order of the conditions within a run was pseudo-randomized to avoid effects of ordering [29]. Each session took approximately 1–1.5hrs and the participant was able to take an optional short break in between runs with a mandatory 1-minute break after the 5th run. The subject was informed of the progress in percentage by a pop-up window after each even numbered run (i.e., 2nd, 4th, 6th, and 8th run).

## Statistical analysis

Our main outcome measures consisted of the mean simulated H-reflex amplitude (performance, $h_{sim}(k)$) and the mean change in dial position between trials (operant strategy, $\Delta\theta(k)$). Operant strategy is a measure of the participant's exploration in finding a hidden target, which was the angular difference between the present dial position and previous trial's dial position. To determine the effects of each of the experimental parameters (feedback type, biological variability, and reward threshold) on the performance and operant strategy of the participants, the average performance and operant strategy of different conditions were compared by one-way repeated measures ANOVA with Tukey HSD post-hoc test ($\alpha < 0.05$). Our main hypotheses were: 1) **Kr** feedback will improve performance compared to **Kp** feedback, 2) larger biological variability will result in worse performance, and 3) the difficult reward threshold (**DT**) will worsen performance and make the operant strategy less aggressive (or exploratory). In addition, to test for interaction effects of experimental parameters on performance and operant strategy (e.g., interaction between the biological variability and feedback type on performance), two-way repeated measures ANOVA with Tukey HSD post-hoc test was used.

## Similarity of learning process during simulated environment: Comparison with real operant H-reflex conditioning

To illustrate similarities between learning in the simulated environment and actual operant H-reflex conditioning behavior, we used a statistical model. We computed a single linear mixed model (LMM) based on conditions C9 and C10 from all participants in the simulated environment (**KpKrLVMT** and **KpKrHVMT**). We included the performance mean across a single run as the dependent variable with the fixed effect of biological variability and the random effect of participant. Using this LMM driven from the simulated environment, we then input measured (i.e., real) H-reflex variability extracted from data collected previously for five healthy and two participants post-stroke performing operant RF H-reflex conditioning [23], to calculate the estimated performance. Later, we normalized the estimated performance magnitude for direct comparison with real operant H-reflex conditioning performance, using the method mentioned in our previous study [23]. We compared the estimated performance from the LMM based on measured H-reflex variability with the real operant H-reflex conditioning performance.

There were 24 training sessions, wherein each session was comprised of 3 runs. Using the model, we predicted the performance mean of each of 72 runs. To examine the prediction accuracy, a Pearson's correlation coefficient with ($df = 70$, $\alpha < 0.05$) was computed to assess the linear relationship between the actual performance and the estimated performance.

## Results

### Effect of biological variability and feedback type on performance

Increased biological variability worsened performance, as evidenced by higher values (**Table 1**). Under both levels of biological variability, the **Kp** feedback type exhibited the poorest performance and the **KpKr** exhibited the best performance (**Kp-KpKr** = *0.037±0.008*, *p < 0.0001, Tukey HSD*) (**F**). The difference was larger during the high variability conditions than low variability conditions. We observed a strong trend towards statistical significance of an interaction effect between the biological variability and feedback type on performance ($F_{(2,216)} = 2.98$, $p = 0.05$, *two-way ANOVA*). A summary of comparisons is provided in **Fig 5** and **Table 1**. Detailed values of performance and comparison are provided in **S1 Table**.

**Table 1. Effect of biological variability and feedback type on performance and operant strategy (moderate threshold).**

| Pair-wise Comparison | Performance | Operant Strategy |
|---|---|---|
| $HV \leftrightarrow LV$ | 0.114±0.004 (***) | -1.449±0.074˚ (ns) |
| $Kp \leftrightarrow KpKr$ | 0.037±0.008 (****) | 1.168±0.080˚ (ns) |
| $Kp \leftrightarrow Kr$ | 0.015±0.005 (*) | 1.512±0.061˚ (ns) |
| $Kr \leftrightarrow KpKr$ | 0.021±0.009 (***) | -0.344±0.086˚ (ns) |

Values represent overall mean ± SE. Tukey HSD post hoc tests conducted for pair-wise comparisons ($\leftrightarrow$) between different conditions.

Statistical significance

(* p<0.05

**p<0.01

***p<0.001

****p<0.0001)

## Effect of biological variability and feedback type on operant strategy

Operant strategy, quantified by the change in angle of the dial ($\Delta\theta$), was not significantly different across biological variability levels. Also, no significant difference in operant strategy was observed across different feedback types overall (**Table 1**). However, during *LV*, operant strategy was more aggressive during *Kr* compared to *Kp* feedback (*Kp-Kr = 4.79 ±0.09, p < 0.0001, Tukey HSD*) (**Fig 5**). This inconsistent result raised the question of whether there was any interaction effect between the biological variability and feedback type on operant strategy. Reduced biological variability enhanced the difference in operant strategy between *Kp* and *Kr* feedback ($F_{(2,216)} = 4.683$, *p < 0.05, two-way ANOVA*). Results are summarized in **Fig 5** and **Table 1**. Detailed values of strategy and comparison are provided in **S1 Table**.

## Effect of biological variability and threshold level on performance

Increased biological variability worsened performance overall (**Table 2**). At low variability, performance was worst at the easy threshold level compared to difficult and moderate threshold levels (**Table 2**). However, at high variability, we did not observe any difference in performance across different threshold levels (p>0.05). Using a two-way repeated measures ANOVA, we observed a significant interaction effect between the biological variability and threshold level ($F_{(2,238)} = 14.2$, *p<0.001, two-way ANOVA*) on performance. A summary of comparisons is found in **Fig 6** and **Table 2**. Detailed values of performance and comparisons are provided in **S2 Table**.

## Effect of biological variability and threshold level on operant strategy

Threshold level had a significant effect on operant strategy ($F_{(2,238)} = 28.61$, *p<0.001, two-way ANOVA*), with an easier threshold resulting in a less aggressive strategy (**Fig 6**). We did not observe a significant effect of biological variability on operant strategy ($F_{(1,238)} = 0.32$, *p = 0.5703, one-way ANOVA*), but increased signal variance decreased the effect of threshold on strategy ($F_{(2,238)} = 6.48$, *p<0.005, two-way ANOVA*) (**Fig 6**). Pairwise comparisons also indicated increased aggressiveness in operant strategy as threshold became more difficult (**Table 2**). Detailed values of strategy and comparisons are provided in **S2 Table**.

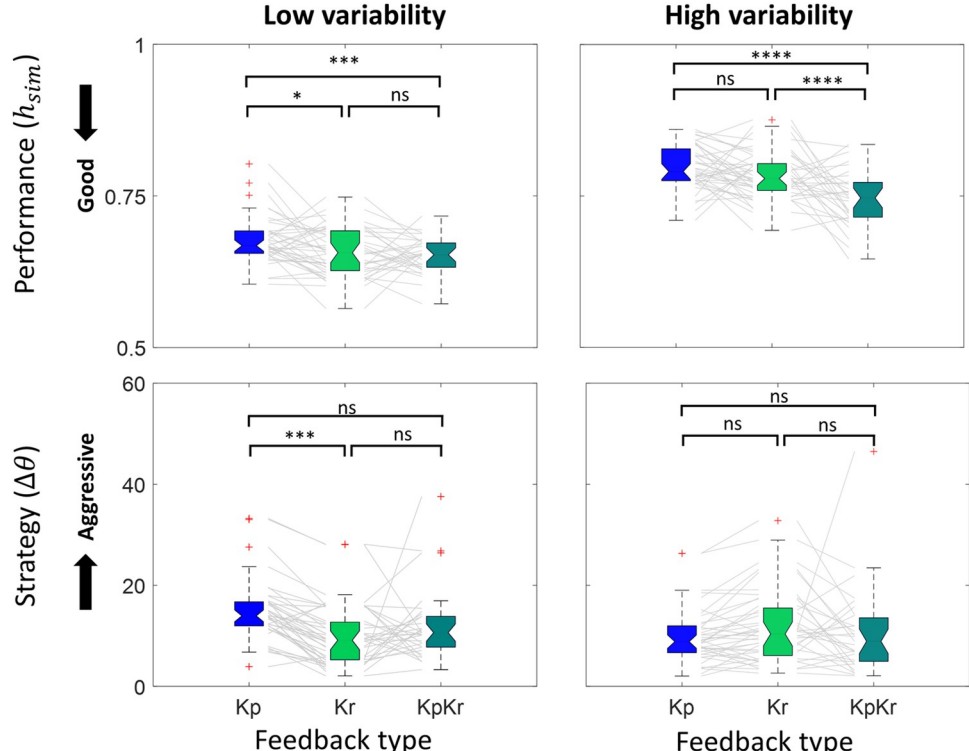

**Fig 5. Effect of biological variability and feedback type on performance and operant strategy (moderate threshold).** On each box, the notch indicates the median, and the bottom and top edges of the box indicate the 25th and 75th percentiles, respectively. The whiskers extend to the most extreme data points not considered outliers. Red cross ('+') marker symbol represents outliers. Grey lines connect each participants' value across different conditions. Each color represents different feedback types (Blue: **Kp**, Green: **Kr**, and Turquoise: **KpKr**). Results for performance and operant strategy under low and high biological variability conditions are shown. Tukey HSD pair-wise comparisons between different feedback type during each biological variability are added. Statistical significance (* p<0.05, **p<0.01, ***p<0.001, ****p<0.0001).

**Table 2. Effect of biological variability and threshold level on performance and operant strategy.**

| Pair-wise comparison | Performance | Operant strategy |
|---|---|---|
| **HV ↔ LV** | 0.105±0.004 (***) | -0.562±0.037 (ns) |
| **DT ↔ ET** | -0.040±0.001 (****) | 8.772±0.071 (****) |
| **DT ↔ MT** | 0.001±0.001 (ns) | 5.281±0.086 (****) |
| **MT ↔ ET** | -0.041±0.001 (****) | 3.485±0.092 (**) |

Values represent overall mean ± SE. Tukey HSD was used for pair-wise comparisons between conditions of high and low variability (**HV, LV**) and difficult, moderate, and easy threshold (**DT, MT, ET**).

Statistical significance

(* p<0.05

**p<0.01

***p<0.001

****p<0.0001)

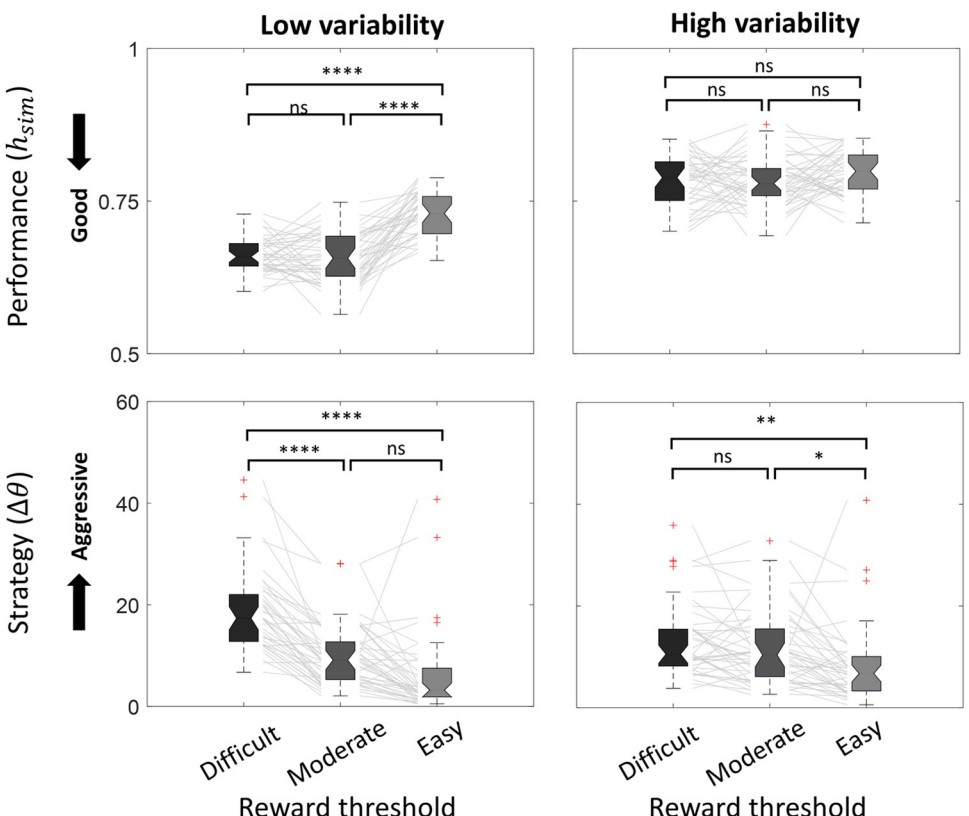

**Fig 6. Effect of biological variability and threshold level on performance and operant strategy.** On each box, the notch indicates the median, and the bottom and top edges of the box indicate the 25th and 75th percentiles, respectively. The whiskers extend to the most extreme data points not considered outliers. Red cross ('+') marker symbol represents outliers. Grey lines connect each participants' value across different conditions. Each color represents different reward thresholds (Black: **Difficult**, Dark gray: **Moderate**, and Light gray: **Easy**). Results under low and high biological variability conditions are exhibited. Tukey HSD was used for pair-wise comparisons between conditions. Statistical significance (* $p<0.05$, **$p<0.01$, ***$p<0.001$, ****$p<0.0001$).

### Similarity in learning during simulation: Comparison with operant H-reflex conditioning result (run-by-run analysis)

Our simulated results demonstrated fair to strong positive correlations with real operant H-reflex conditioning results [30]. Real performance of 7 participants from previous work (5 healthy and 2 post-stroke) [23] was compared to simulated performance estimated by the LMM (**Table 3**). All participants' data were significantly correlated to the simulated data. The strength of the correlations was strong in one healthy individual, moderate in three healthy individuals, and fair in one healthy individual and two post-stroke individuals. A representative example of a strong correlation between real and simulated data is shown in **Fig 7**. Real data (solid line) consisted of the mean performance of each run of operant H-reflex conditioning training (72 runs = 24 sessions x 3 run/session). Estimated performance based on the LMM using the measured H-reflex variability extracted from each specific run is shown as a dashed line (**Fig 7**).

## Discussion

The goal of the present study was to investigate the effect of feedback type, biological variability, and reward threshold on individuals' feedback performances and operant strategies in a

**Table 3. Comparison of the simulated environment to real operant H-reflex conditioning.**

| Subject | Pearson's Correlation | |
|---|---|---|
| | R | p-value |
| H1 | 0.832 | < 0.0001 |
| H2 | 0.676 | < 0.0001 |
| H3 | 0.500 | < 0.0001 |
| H4 | 0.598 | < 0.0001 |
| H5 | 0.491 | 0.0059 |
| S1 | 0.308 | 0.0195 |
| S2 | 0.411 | 0.0003 |

Comparison between two environments comprised of testing a correlation between the performance during real operant H-reflex conditioning for 5 healthy participants (H1-5) and 2 participants with stroke (S1-2) and that of simulated environment.

Statistical significance

(* $p<0.05$

** $p<0.01$

*** $p<0.001$

**** $p<0.0001$)

simulated operant H-reflex conditioning environment. We used a novel simulation environment based on real operant H-reflex conditioning parameters to isolate the effect of feedback parameters on explicit learning. Our main findings, consistent with our hypotheses, were that 1) *Kr* feedback resulted in better performance than *Kp* feedback alone, 2) larger biological variability worsened feedback performance, 3) biological variability modulated the effect of feedback type on operant strategy, 4) biological variability modulated the effect of reward threshold on strategy, and 5) a difficult reward threshold resulted in better performance and more aggressive operant strategies. Performance in the simulated environment, albeit governed entirely by explicit learning processes, was found to be similar to actual operant H-reflex conditioning performance. This study is a new approach to understanding the learning

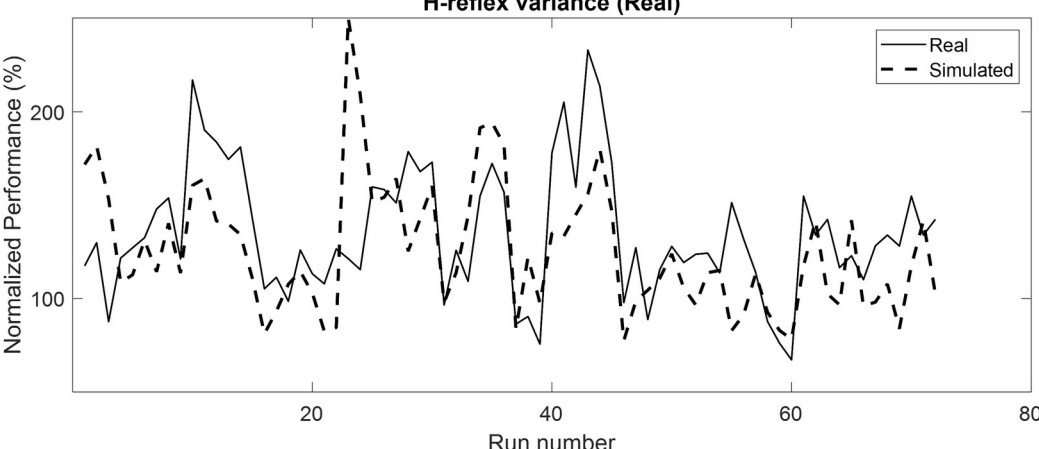

**Fig 7. Representative comparison of real and simulated performance.** Normalized performance (%) of H-reflex for simulated environment, real operant H-reflex conditioning environment, and random environment are presented. A simulated environment driven LMM was used to predict real operant H-reflex performance in a single individual (H1) given the biological variability for each run. The real consisted of the mean performance in each of 72 training runs.

mechanisms of operant H-reflex conditioning. Our results suggest that explicit processes play a role in operant H-reflex conditioning, and this process is modulated by feedback parameters. This paradigm can be used to quickly examine different feedback parameters on operant learning and potentially identify non-responders, ultimately improving the procedural robustness of self-modulation of H-reflex activity.

Neural operant conditioning incorporates both explicit and implicit learning processes that are difficult to delineate. In our earlier work, we used a novel simulated operant conditioning environment approach to separate implicit and explicit processes based on real fMRI data [24] and then validated this model experimentally [25]. One of the novelties of this work is the adaptation of this paradigm for operant H-reflex conditioning, a significant development because it allows investigation of basic skill learning principles. Motor skill learning paradigms have different levels of variability, such as the efferent command and variability of the environmental dynamics [31, 32]. Operant H-reflex conditioning has been classified as a simple motor learning task [33]. In this study, we use an unskilled motor task as a proxy for operant strategy and then simulated the natural variability of the H-reflex [34], thereby transferring efferent variability to environmental variability. This innovation allows investigation into two factors: operant strategy unaffected by efferent noise due to the use of an unskilled task as a proxy, and the controlled analysis of the effect of biological variability.

Thus, this approach enables an understanding of the operant strategies one uses. This perspective into operant strategies is unique compared to tasks such as neurofeedback, where operant strategies can only be reported anecdotally, or motor skill learning, where the explicit strategies can be measured, but are filtered by kinematics and muscle activity.

Biological variability played a multifaceted role on performance and strategy. We expected the performance to be worse with higher biological variability. Signal noise affects motor learning [35, 36], and more specifically, has been shown to reduce operant H-reflex conditioning in animal models [19]. The results from the simulated environment aligned with this expectation. Additionally, we expected one's strategy to be less exploratory or aggressive in case of larger variability given lower reliability on feedback, which was confirmed by our data. We observed that during low variability, *Kp* feedback resulted in more aggressive operant strategies than *Kr* feedback, but we did not observe a difference under high variability. Thus, feedback type affects operant strategy only when the variability is low. This observation is novel because we have focused on the interaction effect of biological variability and feedback type on operant strategy, while other studies have only focused on either the biological variability [32, 37] or feedback type [38].

We have also explored the effect of reward threshold level on performance and operant strategy. Previous work in visuomotor learning [39] has emphasized the effect of negative feedback (e.g., failure) on increased variability of subject's motor strategy during reward-based learning. Also, work in motor learning has indicated that reward threshold affects motor learning [32], and as such, we hypothesized that making the reward threshold more difficult than what is typically provided in operant H-reflex conditioning testing [14] would worsen performance. We observed that performance was affected by threshold level, biological variability, and the interaction between the two. Specifically, at low variability, the easy threshold level worsened performance, however, at high variability, we did not observe a change in performance. Such an interaction is expected as the environment (noise) is known to modulate the effect of task difficulty on motor learning [40]. Given the specific parameters of this task, it would be difficult to compare other work in relation to these findings. However, from an explicit learning perspective, an easier threshold is advised against, when signal noise is sufficiently low.

We observed that at low variability *Kr* feedback was more effective in improving performance than during *Kp* feedback. *Kr* feedback has typically been associated with skilled motor learning processes, in short term and long-term learning [41]. In this study, we show the value of *Kr* in an unskilled task, independent of any long-term learning. Interestingly, when biological variability was high adding *Kp* to *Kr* resulted in further improvement in performance however not when variability was low. Since *Kp* provides feedback of the distance to target, given low variability, and thus high confidence in the accuracy of the feedback, we would expect *Kp* to outperform *Kr*. Within the application of operant reflex conditioning, *Kr* feedback is sufficient for explicit learning of the task. Given the association of *Kr* feedback with long-term learning [41, 42] it likely also suffices for implicit processes. However, across biological variability, *KpKr* feedback showed the best performance. Thus, our results support the continued use of *KpKr* feedback in operant reflex conditioning.

We found fair to strong correlations between performance in our explicit task and real operant H-reflex conditioning. The amount of variance explained for real operant H-reflex conditioning performance by this purely explicit model suggests that explicit learning likely plays a role in operant H-reflex conditioning. Previous studies [33, 43] have suggested that the process of learning or skill acquisition, such as operant conditioning, involves both explicit and implicit mechanisms. Our simulated environment, however, focused only on the explicit mechanisms (i.e., virtual rotary knob control). Given the growing evidence for the primary role of implicit mechanisms in neural operant conditioning [20, 44, 45], this finding that explicit processes explain so much variance in the data was surprising. Although inconclusive, we observed that the biological variability explained by our model was lowest in the two post-stroke individuals. Despite the difference in correlations with our model, the performance of the two post-stroke individuals during real operant H-reflex conditioning was equal or better than the healthy individuals, perhaps suggesting different learning processes occurring post-stroke.

Our simulation paradigm possesses two major limitations that preclude drawing a direct inference of how learning from simulation may be transferred to real operant H-reflex conditioning capability. First, our simulation model was designed as a one-dimensional study, which solely focused on the angle of the rotary knob with a single global maxima and minima. In the real operant H-reflex conditioning environment, however, reported operant strategy is multi-dimensional and varies within and across participants, which could lead to strategic local maxima and minima [14, 46]. The current paradigm should not show run-by-run learning (Fig 7) because there was no strategy or information in this unskilled task to carry over to the following runs. However, future investigations could examine learning on a trial-by-trial basis [47, 48]. Other approaches could adapt this paradigm to incorporate multi-dimensional strategies. Second, our feedback parameters (e.g., biological variability, reward threshold) were chosen based on a limited pool of 7 subjects' data from our prior work [23]. As H-reflex is known for its large biological variability originating from various reasons [49], our two-level variability analysis can be expanded to further investigate the effect of multi-level biological variability.

## Conclusions

We developed a simulated environment of operant H-reflex conditioning to investigate the effects of feedback parameters on explicit learning. The model explained a large portion of the variance of the real H-reflex conditioning despite lacking any implicit learning process. This simulated paradigm may potentially allow the investigation of parameters beyond the capability of real operant H-reflex conditioning and far more efficiently. Our model suggests that the

conditions for best operant H-reflex conditioning performance is to provide both reward and error feedback with at least a moderate (66th percentile success rate) threshold, particularly with low biological variability of the H-reflex. Operant strategy is most aggressive at low variability with error feedback (knowledge of performance) and with a more difficult reward threshold. As variance increases, the effects of feedback type and reward threshold are not as strong. The future of this simulation paradigm may provide a better understanding of learning strategies and ability to identify non-responders, ultimately enhancing the robustness of operant H-reflex conditioning.

## Supporting information

**S1 Table. Effect of biological variability and feedback type on performance and operant strategy (moderate threshold).** Values represent overall mean ± SE. Tukey HSD was used for pair-wise comparisons (↔) between conditions of high and low variability (*HV, LV*) and different feedback types (*Kp, Kr, KpKr*). Statistical significance (* $p < 0.05$, **$p < 0.01$, ***$p < 0.001$, ****$p < 0.0001$).
(PDF)

**S2 Table. Effect of biological variability and reward threshold on performance and operant strategy.** Values represent overall mean ± SE. Tukey HSD was used for pair-wise comparisons (↔) between conditions of high and low variability (*HV, LV*) and difficult, moderate, and easy threshold (*DT, MT, ET*). Statistical significance (* $p < 0.05$, **$p < 0.01$, ***$p < 0.001$, ****$p < 0.0001$).
(PDF)

## Author Contributions

**Conceptualization:** Kyoungsoon Kim, Ethan Oblak, James Sulzer.

**Data curation:** Kyoungsoon Kim.

**Formal analysis:** Kyoungsoon Kim.

**Funding acquisition:** James Sulzer.

**Investigation:** Kyoungsoon Kim, Ethan Oblak.

**Methodology:** Kyoungsoon Kim, Kathleen Manella, James Sulzer.

**Project administration:** James Sulzer.

**Resources:** James Sulzer.

**Software:** Kyoungsoon Kim.

**Supervision:** James Sulzer.

**Validation:** Kyoungsoon Kim, Kathleen Manella.

**Visualization:** Ethan Oblak.

**Writing – original draft:** Kyoungsoon Kim, Kathleen Manella, James Sulzer.

**Writing – review & editing:** Kyoungsoon Kim, Ethan Oblak, Kathleen Manella, James Sulzer.

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
