## [Decision Letter · Decision Letter 0]

25 Sep 2023

PONE-D-23-20613OPERANT REFLEX CONDITIONING SIMULATION ENVIRONMENT REVEALS EFFECTS OF FEEDBACK PARAMETERSPLOS ONE

Dear Dr. Sulzer,

Thank you for submitting your manuscript to PLOS ONE. After careful consideration, we feel that it has merit but does not fully meet PLOS ONE’s publication criteria as it currently stands. Therefore, we invite you to submit a revised version of the manuscript that addresses the points raised during the review process.

Based on the reviewers comments and feedback, "major revision is recommended and re-submit". For detailed comments to Authors, please refer reviewers' feedback section (appended below).

We look forward to receiving your revised manuscript.

Kind regards,

Umer Asgher, PhD

Academic Editor

PLOS ONE

6. We note that Figures 1 and 3 in your submission contain copyrighted images. All PLOS content is published under the Creative Commons Attribution License (CC BY 4.0), which means that the manuscript, images, and Supporting Information files will be freely available online, and any third party is permitted to access, download, copy, distribute, and use these materials in any way, even commercially, with proper attribution. For more information, see our copyright guidelines: http://journals.plos.org/plosone/s/licenses-and-copyright.

1. You may seek permission from the original copyright holder of Figures 1 and 3 to publish the content specifically under the CC BY 4.0 license.

Reviewers' comments:

Reviewer's Responses to Questions

**Comments to the Author**

1. Is the manuscript technically sound, and do the data support the conclusions?

Reviewer #1: Yes

Reviewer #2: Yes

2. Has the statistical analysis been performed appropriately and rigorously? 

Reviewer #1: Yes

Reviewer #2: Yes

3. Have the authors made all data underlying the findings in their manuscript fully available?

Reviewer #1: Yes

Reviewer #2: Yes

4. Is the manuscript presented in an intelligible fashion and written in standard English?

Reviewer #1: Yes

Reviewer #2: Yes

5. Review Comments to the Author

Reviewer #1: I will be evaluating this research based on PLOS ONE predefined criteria. Below is my review for “OPERANT REFLEX CONDITIONING SIMULATION ENVIRONMENT REVEALS EFFECTS OF FEEDBACK PARAMETERS”.

1. The study presents the results of original research.

Yes. To my knowledge there is no prior research that demonstrates the impact of feedback type, signal quality and success threshold on performance and strategy of a visuomotor task. However, there are several studies that have looked at these types of tasks and the strategy used across learner types within both implicit and explicit domains. It would improve the scholarship and emphasis on originality of this research by including the following references.

Brooks, V., Hipperath, F., Brooks, M., Ross, H., & Freund, H. (1995). Learning What and How in a Human Motor Task. Learning & Memory, 2(5), 225–243. https://doi.org/10.1101/lm.2.5.225

Hooyman, A., Gordon, J., & Winstein, C. (2021). Unique behavioral strategies in visuomotor learning: Hope for the non-learner. Human Movement Science, 79, 102858. https://doi.org/10.1016/j.humov.2021.102858

Holland, P., Codol, O., & Galea, J. M. (2018). Contribution of explicit processes to reinforcement-based motor learning. Journal of Neurophysiology, 119(6), 2241–2255. https://doi.org/10.1152/jn.00901.2017

Each paper identified “non-learners” and how the use, or lack thereof, of explicit and implicit strategies drives performance.

2. Results reported have not been published elsewhere.

I believe this to be correct, with the exception of a preprint in bioRxiv.

3. Experiments, statistics, and other analyses are performed to a high technical standard and are described in sufficient detail.

I am uncertain of how the learning of this task is equivalent to learning how to modulate actual H-reflex. Is there prior evidence that shows how learning capability of this task is related to individual capability to change H-reflex? Clarification on how learning of this task is related to learning of H-reflex is needed. What is lost if the simulated H-reflex is removed and every step after decision gain in figure 1 is maintained. What difference in visual feedback would this create that requires the simulated H-reflex to be included? I do recognize that you compare the virtual performance here to that of real performance of H-reflex conditioning from previously collected data [23]. However, this would at best represent an association between these types of learning and not necessarily a transfer of learning.

What is the y axis of figure 7? In the mixed effect model did you also include trial number as a fixed effect? Is the outcome variable for the trained LMM the same as hsim? Do you train an LMM on the virtual reflex data and then test it on the real data? I am having difficulty translating your stat method to figure 7. I thought hsim ranged from 0 to 1 but figure 7 shows an outcome variable ranging from 50 to 250? Also, for the fixed effect of variability, is that the biological variability or some other form of variability? Typically, in validation, although correlation is acceptable, other metrics of validation are MAE, MSE and RMSE. Additionally, you may consider replacing figure 7 with a Bland-Altman plot to demonstrate the agreement between real and simulated. Lastly, I think you need to provide some threshold by which the model is validated or not. To have a strong correlation in one out of 7 test participants isn’t strong evidence of validation. Alternatively, you could generate random data to be compared to the validated data and perform a contrast between the error in predictive accuracy.

Please provide a little more context for how to interpret hsim, the primary performance outcome measure. Please provide an example of what an hsim of 1 versus an hsim of 0 represents.

Could you please provide a visual of the Kr versus Kp versus KpKr display that participants would see under each condition?

Can you please include individual dots in the bar graphs to represent individual participant performance. It would also be good to use boxplots instead of bar graphs.

In the results please provide model estimates and confidence intervals of performance across conditions and interactions. You provide pairwise comparisons but condition performance should also be reported.

Although you demonstrate that mean group performance among the LV/KP condition had the worse performance and was the most aggressive this doesn’t really tell us if greater aggression/exploration is related to worse performance or if this is the case across all conditions. A similar situation appears in the reward threshold to variability result as well. IT would be helpful to see scatter plots of aggression versus performance stratified by group to better understand if these data support hypothesis 3: “the difficult reward criterion will worsen performance and make operant strategy less aggressive”.

4. Conclusions are presented in an appropriate fashion and are supported by the data.

I think the conclusions are presented well but I am not sure that results from this study indicate that explicit processes play a role in operant H-reflex conditioning, and this process is modulated by feedback parameters. I think the results the role of explicit processes on a visuomotor skill and how they are modified due to feedback parameters and feedback noise.

5. The article is presented in an intelligible fashion and is written in standard English.

Yes.

6. The research meets all applicable standards for the ethics of experimentation and research integrity.

Yes. This experiment examined visuomotor strategy among young, non-disabled adults. All participants knowingly gave consent to participate in this experiment.

7. The article adheres to appropriate reporting guidelines and community standards for data availability.

Yes. The data are available on open science framework.

Reviewer #2: Summary:

The manuscript discusses operant conditioning of neural activation and the role of feedback in shaping explicit decision-making processes. The authors developed a simulated operant conditioning environment and conducted an experiment involving 41 participants. They examined the impact of feedback type, signal quality, success threshold, and biological variability on operant conditioning performance and strategy. The study found that performance was influenced by variability, while operant strategy was affected by feedback type.

Review:

The manuscript explores an interesting and relevant topic in the field of operant conditioning and neural activation.

1. The introduction lacks a clear explanation of the significance of the research and its potential applications.

2. While the research aims to determine the explicit decision-making processes in response to feedback, the specific research questions or hypotheses are not explicitly stated in the introduction. It would be useful to outline the research objectives clearly.

3. The manuscript briefly describes the experimental setup but lacks sufficient detail about the web application game and the operant conditioning model. A more comprehensive description of the methodology, including the design of the game and how feedback was provided, would enhance the paper's clarity.

4. The paper mentions extracting parameters from real operant conditioning data but does not provide details about the statistical methods or analysis techniques used.

5. The authors should delve deeper into the significance of the observed relationships between feedback parameters and provide insights into how these findings can be applied or expanded upon. The manuscript should address the limitations of the study and suggest avenues for future research.

6. Consider including papers using dynamic activation in regards to neural networks. This will help make the manuscript take a more round shape regarding the literature review. Try to add some relevant papers from the PLOS Journal.

Here are a couple of references that might be helpful.

Rane, Chinmay, Kanishka Tyagi, and Michael Manry. "Optimizing Performance of Feedforward and Convolutional Neural Networks through Dynamic Activation Functions." arXiv preprint arXiv:2308.05724 (2023).

Biswas, Koushik, Sandeep Kumar, Shilpak Banerjee, and Ashish Kumar Pandey. "TanhSoft—dynamic trainable activation functions for faster learning and better performance." IEEE Access 9 (2021): 120613-120623.

Karthikeyan, Anitha, Ashokkumar Srinivasan, Sundaram Arun, and Karthikeyan Rajagopal. "Complex network dynamics of a memristor neuron model with piecewise linear activation function." The European Physical Journal Special Topics 231, no. 22-23 (2022): 4089-4096.

6. PLOS authors have the option to publish the peer review history of their article (what does this mean?). If published, this will include your full peer review and any attached files.

Reviewer #1: No

Reviewer #2: **Yes: **Kanishka Tyagi

---

## [Author Response · Author response to Decision Letter 0]

15 Dec 2023

All responses to reviewers are included in the attached response to reviewers document.

---

## [Decision Letter · Decision Letter 1]

8 Feb 2024

PONE-D-23-20613R1Simulated operant reflex conditioning environment reveals effects of feedback parametersPLOS ONE

Dear Dr. Sulzer,

Thank you for submitting your manuscript to PLOS ONE. After careful consideration, we feel that it has merit but does not fully meet PLOS ONE’s publication criteria as it currently stands. Therefore, we invite you to submit a revised version of the manuscript that addresses the points raised during the review process.

The original two referees have carefully reviewed the revised manuscript entitled, “Simulated operant reflex conditioning environment reveals effects of feedback parameters". Their comments are appended below. The second reviewer is satisfied the revision, while the first reviewer still has some minor concerns which should be considered before publication. This Academic Editor is sure the critical concerns make the manuscript strengthen. I will consider after receiving the revised manuscript with your replies to each comment.

We look forward to receiving your revised manuscript.

Kind regards,

Manabu Sakakibara, Ph.D.

Academic Editor

PLOS ONE

Journal Requirements:

Reviewers' comments:

Reviewer's Responses to Questions

**Comments to the Author**

1. If the authors have adequately addressed your comments raised in a previous round of review and you feel that this manuscript is now acceptable for publication, you may indicate that here to bypass the “Comments to the Author” section, enter your conflict of interest statement in the “Confidential to Editor” section, and submit your "Accept" recommendation.

Reviewer #1: (No Response)

Reviewer #2: All comments have been addressed

2. Is the manuscript technically sound, and do the data support the conclusions?

Reviewer #1: Partly

Reviewer #2: Yes

3. Has the statistical analysis been performed appropriately and rigorously? 

Reviewer #1: I Don't Know

Reviewer #2: Yes

4. Have the authors made all data underlying the findings in their manuscript fully available?

Reviewer #1: No

Reviewer #2: No

5. Is the manuscript presented in an intelligible fashion and written in standard English?

Reviewer #1: Yes

Reviewer #2: Yes

6. Review Comments to the Author

Reviewer #1: Thank you very much for the thorough and considerate response to my initial critique.

I only have a few minor recommendations before providing a decision of accept.

I am unsure if the data with your OSF directory are finalized but as I look at them in their current form they are nearly impossible to translate back to the experimental paradigm.

From what I can gather, each .txt file consists of the 10 conditions (which were presented in a random sequence?) with each condition consisting of 35 trials? I cannot tell which data point corresponds to which trial (or run) or condition and I believe that the three files for each participant represents the separate sessions? But then each column within each data file has no variable name. Even as I try to carefully read this manuscript, the data in the OSF only look like random numbers. Even when I try to plot them they just look like noise. These data either need a readme file and a complete reformatting to allow outside users to understand what they are looking at. Also, the data should not just be released as individual files. There should just be one large parent file with a column for participant ID, session, and whatever strtg means in the file name.

My motivation for this is because I wanted to look at the data as a response to my initial comment,

“In the mixed effect model did you also include trial number as a fixed effect?”

You responded that run number showed no effect on the model. This is surprising given this is a study on learning. There being no effect of run or trial number would indicate that participants do not improve overtime and thus no learning is actually taking place? When I look at fig 7, now with a y-axis label, I see now that their doesn’t appear to be any change in performance across the several runs, or at least it is impossible to infer from this graph given there is so much nested within each run (10 conditions presented at random, with 35 trials per condition). What is the explanation for this?

Reviewer #2: The author have diligently addressed all the changes and recommendations provided in the previous review. The revised paper is now fully prepared for submission. Thank you for taking all the constructive feedback and guidance.

7. PLOS authors have the option to publish the peer review history of their article (what does this mean?). If published, this will include your full peer review and any attached files.

Reviewer #1: No

Reviewer #2: **Yes: **Kanishka Tyagi

---

## [Author Response · Author response to Decision Letter 1]

21 Feb 2024

We thank the editor and reviewers for their valuable comments that have strengthened the manuscript. We have carefully reviewed these comments and we believe they have been sufficiently addressed. We have primarily addressed the comments in the manuscript but have included excerpts in this document where we felt necessary. Our responses are in red font. 

The line numbers refer to lines in ‘Track Changes (Simple Markup)’ mode. Thank you very much for your time, effort, and attention.

Reviewer Comments:

Reviewer 1

1. I only have a few minor recommendations before providing a decision of accept.

I am unsure if the data with your OSF directory are finalized but as I look at them in their current form they are nearly impossible to translate back to the experimental paradigm.From what I can gather, each .txt file consists of the 10 conditions (which were presented in a random sequence?) with each condition consisting of 35 trials? I cannot tell which data point corresponds to which trial (or run) or condition and I believe that the three files for each participant represents the separate sessions? But then each column within each data file has no variable name. Even as I try to carefully read this manuscript, the data in the OSF only look like random numbers. Even when I try to plot them they just look like noise. These data either need a readme file and a complete reformatting to allow outside users to understand what they are looking at. Also, the data should not just be released as individual files. There should just be one large parent file with a column for participant ID, session, and whatever strtg means in the file name.

>> Thank you for the detailed and valuable comment. We have added a readme.txt file to the osf dataset that explains the data nomenclature.

2. My motivation for this is because I wanted to look at the data as a response to my initial comment,

“In the mixed effect model did you also include trial number as a fixed effect?”

You responded that run number showed no effect on the model. This is surprising given this is a study on learning. There being no effect of run or trial number would indicate that participants do not improve overtime and thus no learning is actually taking place? When I look at fig 7, now with a y-axis label, I see now that their doesn’t appear to be any change in performance across the several runs, or at least it is impossible to infer from this graph given there is so much nested within each run (10 conditions presented at random, with 35 trials per condition). What is the explanation for this?

>> Thank you for the valuable comment. We understand the confusion because in a typical motor learning paradigm there is run-by-run learning. However, this paradigm is an unskilled task, where no useful information is transferred to the next run. For instance, the target location randomly varies by run, and the variation of strategy to reach the target is not a difficult procedure, so any learning would show a ceiling effect quickly. We did not predict any learning between runs, and did not include the statistical analysis because it would be an unnecessary test. However, as could be inferred from Figure 7, there was no learning between runs based on an LMM we ran post-hoc. We have succinctly addressed the reviewer’s concern in the limitations paragraph:

Lines 503-6:

The current paradigm should not show run-by-run learning (Figure 7) because there was no strategy or information in this unskilled task to carry over to the following runs. However, future investigations could examine learning on a trial-by-trial basis [47, 48].

---

## [Decision Letter · Decision Letter 2]

27 Feb 2024

Simulated operant reflex conditioning environment reveals effects of feedback parameters

PONE-D-23-20613R2

Dear Dr. Sulzer,

We’re pleased to inform you that your manuscript has been judged scientifically suitable for publication and will be formally accepted for publication once it meets all outstanding technical requirements.

Kind regards,

Manabu Sakakibara, Ph.D.

Academic Editor

PLOS ONE

Additional Editor Comments (optional):

Reviewers' comments:

Reviewer's Responses to Questions

**Comments to the Author**

1. If the authors have adequately addressed your comments raised in a previous round of review and you feel that this manuscript is now acceptable for publication, you may indicate that here to bypass the “Comments to the Author” section, enter your conflict of interest statement in the “Confidential to Editor” section, and submit your "Accept" recommendation.

Reviewer #1: All comments have been addressed

2. Is the manuscript technically sound, and do the data support the conclusions?

Reviewer #1: Yes

3. Has the statistical analysis been performed appropriately and rigorously? 

Reviewer #1: Yes

4. Have the authors made all data underlying the findings in their manuscript fully available?

Reviewer #1: Yes

5. Is the manuscript presented in an intelligible fashion and written in standard English?

Reviewer #1: Yes

6. Review Comments to the Author

Reviewer #1: Thank you very much for conscientiously responding to each of prior comments. I hope this article can have a meaningful impact on the field.

7. PLOS authors have the option to publish the peer review history of their article (what does this mean?). If published, this will include your full peer review and any attached files.

Reviewer #1: No

---

## [Editor Report · Acceptance letter]

12 Mar 2024

PONE-D-23-20613R2 

PLOS ONE

Dear Dr. Sulzer, 

I'm pleased to inform you that your manuscript has been deemed suitable for publication in PLOS ONE. Congratulations! Your manuscript is now being handed over to our production team.

Kind regards, 

on behalf of

Dr. Manabu Sakakibara 

Academic Editor

PLOS ONE